# A Bio-inspired Gradient-Free Learning Framework for CPG-based Neural Networks in Robot Locomotion

## Abstract

Central Pattern Generators (CPGs) for animal locomotion have been adopted for robot locomotion control for three decades, but have rarely been used in reinforcement learning. It is partly due to the inherent recursive connections causing gradient vanishment or explosion during optimization. In this paper, we propose a framework with a bio-inspired learning rule to optimize a recurrent neural network for robot reinforcement learning. The learning rule utilizes weight fluctuation for parameter exploration and reward for convergence. With the framework, we trained a minimal network containing CPG and full connection layers for quadrupedal locomotion. Experiments suggest that our framework can train the network, while the Proximal Policy Optimization (PPO) fails. Compared to PPO with a feedforward network, our framework with a CPG is more robust to sensor failure. This work lowers the barrier to exploring the potential advantages of recurrent neural networks in robot reinforcement learning.

## 1 Introduction

Legged robots have attracted significant attention over the last decades (Bar-Cohen, 2006; Pfeifer et al., 2007), and the need for robust locomotion capabilities arises. Traditional control methods based on numerical, kinematic, and geometric approaches (Ostrowski & Burdick, 1996) do not fulfill these requirements. Although quadruped control using Reinforcement Learning (RL) has been greatly developed (Iscen et al., 2018; Lee et al., 2020; Ji et al., 2022; Yang et al., 2022), it often requires additional designed phases for periodic behaviors, causing increased complexity and training costs.

To address the challenge, researchers have focused on bio-inspired approaches. Central Pattern Generator (CPG), a biological neural circuit capable of producing rhythmic signals without rhythmic input (Brown, 1911), gradually attracts attention. CPGs are prevalent in both vertebrates (Hultborn & Nielsen, 2007; Guertin, 2009; Danner et al., 2015) and invertebrates (Orlovsky et al., 1999), driving stereotyped motor behaviors such as walking, swimming, respiration and mastication (Ijspeert, 2008; Yu et al., 2013). CPGs consist of interconnected inhibitory/excitatory circuits, generate rhythmic patterns autonomously, and can be modulated by external signals (e.g., sensory feedback) (Nadim & Bucher, 2014) to adjust global behavioral modes.

CPG-based control has proven effective for legged robot locomotion (Taga et al., 1991; Liu et al., 2011; Wang et al., 2012). In RL, initial values of parameters significantly influence the training speed and results, while CPGs can provide advantageous initial values for periodic tasks, reducing model complexity and computational demands. Therefore, it is feasible to integrate CPGs into RL frameworks, enabling robust motion control (Fukunaga et al., 2004; Nakamura et al., 2007; Amrollah & Henaff, 2010; Cho et al., 2019; Bellegarda & Ijspeert, 2022). Our experiments demonstrate that the CPG-based networks maintain stable performance even without feedback, exhibiting strong robustness (Fig.6).

However, the recurrent connectivity of CPGs makes it difficult to directly optimize parameters via gradient descent based on error backpropagation. Existing solutions include trial-and-error methods and optimization-based approaches (Ijspeert, 2008). Trial-and-error relies on iterative experimentation to identify task-specific parameters, such as tactile interaction-based tuning (Dalla Libera et al.,

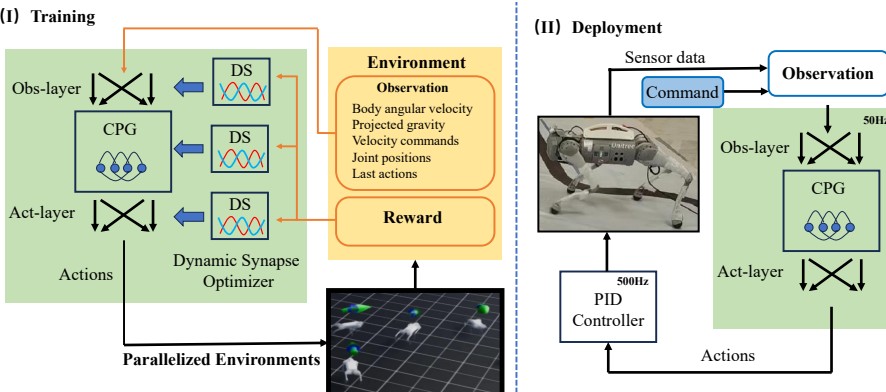

Figure 1: The structure of our model. The observation is fed into the CPG as input through a linear layer, called the Obs-layer. The output of the CPG is fed into the PID controller through another linear layer, called the Act-layer. Obs-layer, CPG, and Act-layer are all equipped with a dynamic synaptic optimizer.

2010), or analytical design of CPGs for smooth gait transitions (Liu et al., 0), but these methods depend on researchers' expertise and do not support learning of the predefined weights, causing high temporal cost.

General optimization approaches are also limited. Truncated backpropagation through time (Sato, 1990) preserves historical state trajectories to optimize CPG parameters, incurring high computational and storage costs. Campanaro et al. (2021); Yang et al. (2025) addressed CPG recursiveness by exposing internal states and expanding them into differentiable stateless networks, but the internal state and recursive complexity of CPG leads to larger models and higher computational demands. Other approaches optimize CPGs separately using Evolutionary Strategy (ES) (Shi et al., 2022) or Genetic Algorithm (GA) (Wang et al., 2021), while training the rest network via gradient descent, requiring additional design effort and training overhead. In addition, the CPG-actor-critic framework (Nakamura et al., 2007) treats the CPG, robot, and environment as a coupled dynamic system, but relies on fixed internal CPG weights, limiting motion diversity.

Recent advances in observational techniques have enabled detailed characterization of neuronal structures and synaptic dynamics (Son et al., 2024). Biologically, CPG parameter regulation primarily corresponds to the changes in synaptic strength, which mainly depend on the quantity and spatial distribution of transmitter receptors within synapses (Groc & Choquet, 2020). Existing studies revealed that synaptic volume and protein density exhibit spontaneous, periodic fluctuations over timescales from minutes to days, independent of external stimuli and not strictly unidirectional (Choquet & Triller, 2013; Cizeron et al., 2020; Son et al., 2024). These fluctuations arise from lateral movements of transmitter receptors and their transport via endocytosis/exocytosis, and thus cause unstable synaptic strength even without learning.

Building on these insights, novel non-steady-state synaptic models have been proposed to explain learning in biological neural circuits (Wei & Webb, 2018b). By modeling nonlinear dynamics of transmitter receptor motion, spontaneous chaotic oscillations emerge, traversing regions covered by strange attractors. Simple learning tasks can be achieved by establishing linear update rules between synaptic volume, receptor motion damping, and neuromodulators. For computational efficiency and parameter robustness, this mechanism can be further simplified using periodic sinusoid functions, which is applied in reinforcement learning tasks (Wei & Webb, 2018a). Thus, the methods enable co-training of CPGs with neural networks while avoiding gradient vanishing/exploding issues. However, prior implementations lack compatibility with mainstream RL training and parallel acceleration, resulting in low training efficiency.

In this work, we introduce a dynamic synaptic optimizer compatible with general neural network training and parallel learning, enabling efficient GPU-accelerated RL for complex tasks. For the locomotion task, we construct a three-layer network with a neural oscillator-based CPG (Fig.1) and benchmark the optimizer against PPO. Results suggest that our bio-inspired mechanism dynamically

modulates CPG parameters, directly optimizing the network and providing a simple, low-cost solution for online CPG training in RL. The CPG-based network exhibits robustness to sensor failures compared to PPO-trained feedforward networks.

## 2 METHODS

There are two set of works involved in this paper: a Dynamic Synapse (DS) optimizer for optimization free from gradient descent and a hybrid network containing Multilayer Perception (MLP) and CPGs for robot locomotion. 1) DS optimizer represents a simplification of synaptic learning processes in neural circuits, which can optimize the hybrid network directly. We modulate two critical variables (Eq.3) as function of the reward (which could be positive or negative) to influence weights fluctuation dynamics so that weights can be maintained within high-reward regions of weight space, which enables network learning. 2) The hybrid network plays a role of gait generation, which is centered on a CPG consists of Matsuoka neural oscillators (Matsuoka, 1985) or FitzHugh-Nagumo neural oscillators (FitzHugh, 1961; Nagumo et al., 1962). Under appropriate parameter configurations, this CPG autonomously generates rhythmic signals without external inputs. The network contains fully connected layers for modulation of signal inputs and action outputs, which are trained in a certain order, either alone or together with the CPG, and thus produce stable quadrupedal gaits.

### 2.1 DYNAMIC SYNAPSE OPTIMIZER

In biological systems, synaptic learning can be achieved by neuromodulators regulating the movement of transmitter receptors dynamics at the postsynaptic membrane. However, modeling complete biochemical cascade reactions results in extra computational costs and lower training efficiency. In previous works(Wei & Webb, 2018a;b), the mechanism has been simplified to a dynamic synapse model that has been proven to facilitate reinforcement learning. Based on it, we design a PyTorch-based optimizer with parallel training support, improving compatibility and learning efficiency compared to the original version.

#### 2.1.1 WEIGHTS FLUCTUATION

The DS models synaptic strength as the concentration of the receptor within the synapse, and positive and negative feedback during receptor transport across the concentration gradient are also considered to model the dynamics. Here, to simplify the calculations, we directly use a period-stochastic sinusoid function to represent the oscillations of synaptic strength, i.e., network weights. For a single synapse during the $i$th period, its strength varies with time t:

$$W(t) = A\sin(2\pi\frac{t - \sum_{k=0}^{i-1}T_k}{T_i}) + C, \text{if} \sum_{k=0}^{i-1}T_k \le t \le \sum_{k=0}^{i}T_k \tag{1}$$

where $W$ is instantaneous weight, $A$ the amplitude of fluctuation, $T_k$ the $k$th period and $C$ controls the centre of fluctuation.

After the weight fluctuates a complete period(crosses its current centre from the same direction), a new period is sampled from a normal distribution:

$$T_i \sim \mathcal{N}(\mu, \sigma^2) \tag{2}$$

where $\mu$ stands for mean of the distribution while $\sigma$ stands for variance. Although Wei & Webb (2018a) treated $\mu$ as a hyperparameter, in our optimizer $\sigma$ is proportional to $\mu$ to enhance the randomness of period generation for better exploration of weight.

In practical experiments, $A$,$C$ and $\mu$ are predefined with initial values based on task requirements, then iteratively updated together with other parameters according to the Eq.3. Note that every weight fluctuation is independent to avoid periodic exploration (Fig. 2 (c) and (d)) and instead helps to sample the weight space densely(Fig. 2 (a) and (b)).

#### 2.1.2 LEARNING RULES

Continuous exploration in the weight space induces reward variations caused by the agent's action (i.e., network outputs). It is evident that the weight dynamics is modulated by the fluctuation amplitude $A$ and fluctuation center $C$. Our goal is to control these two variables to stabilize weights

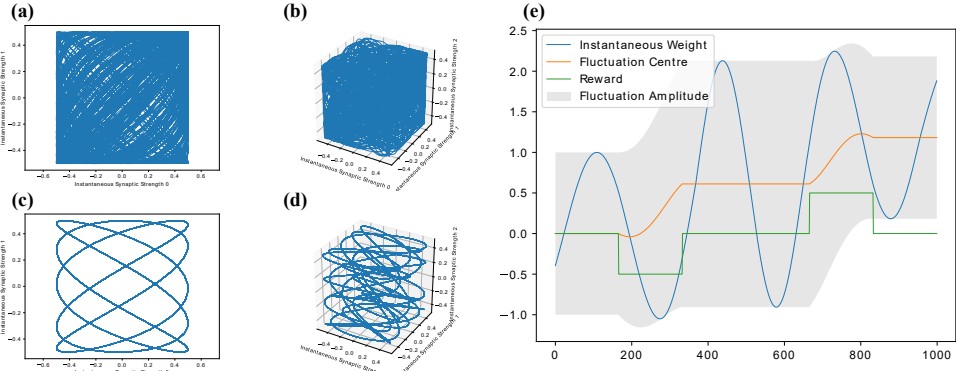

Figure 2: Mechanism of dynamic synapse optimizer. Only independent fluctuations (a) and (b) can enable densely searching; Locked period causes periodic and limited exploration in weight space like (c) and (d). (e) The learning rules of dynamic synapse optimizer. Weight (blue) fluctuates around a center (orange). When a positive reward (green) is received, the center shifts towards instantaneous weight, and the exploration amplitude (shadow) converges. Similarly, the center shifts in the opposite direction when a negative reward is encountered, and the amplitude amplifies.

within regions that maximize rewards while avoiding punishment (i.e., negative rewards) zones. Specifically, 1) Weight center $C$ shifts towards higher rewards regions; 2) Fluctuation amplitude $A$ decreases when positive rewards are obtained and increases when punishments are encountered. This strategy directly correlates weight update dynamics with reward feedback, as mathematically formalized:

$$\dot{C} = \alpha(W(t) - C)R(t)$$
$$\dot{A} = -\beta A R(t)$$

(3)

where $\alpha$ is the learning rate of weight centre, $R$ the reward, $\beta$ the convergence rate of fluctuation amplitude. It can be observed that when the agent's action generates a positive reward, the fluctuation center shifts toward the instantaneous weight $W$ that induces the reward, while the fluctuation amplitude simultaneously decreases to prompt the convergence (Fig.2 (e) positive reward). Similarly, the center shifts in the opposite direction when a negative reward is obtained, while the fluctuation amplitude increases to enhance the exploration in weight space at the same time (Fig.2 (e) negative reward). Consequently, synaptic strength gradually converges within regions capable of yielding higher rewards under continuous modulation.

**Parallel Acceleration**  Prior implementation of the dynamic synapse optimizer has been attempted only on small-scale networks and lacks support for parallel training (Wei & Webb, 2018a), so we refactor it using PyTorch to enhance compatibility and training efficiency. The new implementation leverages multiple rewards generated from parallel training instances with shared weights. Specifically, during weight center updates, instead of single reward obtained from one instance, all the concurrent training instances' rewards are taken into account, i.e., for a single synapse:

$$\dot{C} = \alpha(W(t) - C)\frac{\sum_j^n R_j(t)}{n}$$
$$\dot{A} = -\beta A \frac{\sum_j^n R_j(t)}{n}$$

(4)

where $R_j$ is the reward from $j$th instance in parallel training at moment $t$.

## 2.2 NETWORK

We construct a three-layer network con- taining a neural oscillator-based CPG. The observation is fed into the CPG through a linear layer, called the Obs-layer. The output of the CPG is fed into the PID controller through another linear layer, called the Act-layer (Fig.1).

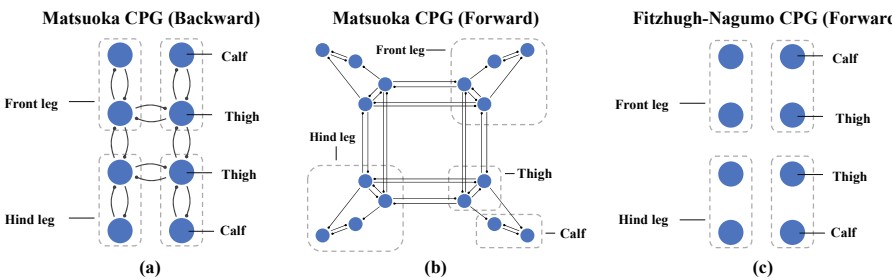

Figure 3: Three kinds of CPGs. Each of the structure corresponds to the robot's four legs one by one (illustrated by grey dashed box). Note that only in Matsuoka CPG for forward walking (b) we use two neurons to compose thigh and calf for each leg.

Three CPG structures were tested separately (Fig.3): the first two employed Matsuoka oscillators (Matsuoka, 1985), while the third utilized FitzHugh-Nagumo (FHN) oscillators (FitzHugh, 1961; Nagumo et al., 1962). It is worth noting that in the third structure, no internal connections were established within the CPG, allowing each neuron to oscillate independently. The equations and explanations for both types of neural oscillators are provided below.

**Matsuoka Oscillator**    This oscillator uses a system of first-order differential equations to control the membrane potential changes of neurons. Each neuron receives positive signals from other neurons and generates a negative self-inhibitory signal. This model simulates the operation process of neurons and has strong stability. The equation of this neuron model is as follows:

$$\tau_r \dot{u}_i = -u_i + \sum_j^n w_{ij} y_j - \beta v_i + I$$
$$\tau_a \dot{v}_i = -v_i + y_i$$
$$y_i = \max(u_i, 0)$$

(5)

where $u$ represents the state of neuron, $v$ is the self-inhibition variable, $y$ is the output of the neuron, and $\tau_r$, $\tau_a$, $\beta$ are parameters that control the dynamics of the system, and they are also learnable parameters. $I$ is the input current, $w_{ij}$ is the weight of the connection from neuron $j$ to neuron $i$, and $n$ is the total number of neurons connected to neuron $i$.

**FitzHugh-Nagumo Oscillator**    This model is a simplified version of the Hodgkin-Huxley model, which describes the electrical activity of excitable cells such as neurons. The FHN model consists of two coupled differential equations that represent the dynamics of the membrane potential and a recovery variable. The equations are as follows:

$$\dot{v} = v - \frac{v^3}{3} - w + I$$
$$\dot{w} = a(bv - cw)$$

(6)

where $v$ is the membrane potential, $w$ is the recovery variable, $I$ is the external input current, and $a$, $b$, and $c$ are parameters that control the dynamics of the system. The FHN model captures the essential features of neuronal excitability and can exhibit various behaviors such as resting state, periodic spiking, and bursting depending on the parameter values and input current.

## 2.3 QUADRUPED ROBOT

The quadruped robot we use in experiments is Unitree go1, which has four legs, and each equipped with three degrees of freedom. On each leg, the hip joint is equipped with two degrees of freedom, and the knee joint is equipped with one degree of freedom. The two joints located in the hip are used to control the forward and backward swing of the leg (pitch), and to control the internal and external rotation (roll), which is used for directional adjustment and gait switching. The joint in the knee is

used to control leg flexion and extension, which is used to adjust stride length and jump height. The robot contains a total of 12 aluminum alloy precision joint motors and encoders, and 4 underfoot plantar sensors with IMUs for joint angle manipulation and environment perception.

## 2.4 OBSERVATION SPACE AND ACTION SPACE.

The observation space has a total of 37 dimensions, including 3-dimensional angular velocity $\omega_a$, 3-dimensional projected gravity $G$, 3-dimensional target command $C$, 12-dimensional joint position $q$, 12-dimensional last action $a_{t-1}$, and 4-dimensional foot contact statuses $c_f$. The target command $C$ includes target linear velocity $v_c$ and target angular velocity $\omega_c$. More details of symbols are provided in the appendix A. The action space consists of 12-dimensional joint target angles. It will be multiplied by the proportional coefficient and added to the joint offset before being sent to the Proportional Integral Derivative (PID) controller.

## 2.5 REWARDS FUNCTION.

In order to enable the dynamic synaptic optimizer to smoothly optimize the model parameters, a series of reward functions (Tab. 1) are designed. $r_0$ and $r_1$ ensure that the robot follows the given linear velocity and angular velocity. $r_2$, $r_3$, and $r_4$ help the robot run more smoothly and stably. $r_5$, $r_6$ and $r_7$ avoid robots that produce excessive output or drastic changes. $r_8$ and $r_8$ encourage the robot to take longer steps and reduce sliding its feet on the ground. $r_{10}$ helps the robot maintain an appropriate trunk height during locomotion, preventing crouched postures.

Table 1: Reward configurations

| Reward | Expression | Weight | Reward | Expression | Weight |
|--------|-----------|--------|--------|-----------|--------|
| $r_0$ | $\exp\left(-\frac{\|v_c-v_{xy}\|_2}{std^2}\right)$ | 1.5 | $r_6$ | $\|\ddot{q}\|_2$ | $-2.5 \times 10^{-8}$ |
| $r_1$ | $\exp\left(-\frac{\|\omega_c-\omega_z\|_2}{std^2}\right)$ | 0.75 | $r_7$ | $\|a_t-a_{t-1}\|_2$ | $10^{-3}$ |
| $r_2$ | $\|v_z\|_2$ | $-0.2$ | $r_8$ | $\sum_{i=1}^{4}(t_{air}-0.5)c_f$ | 1 |
| $r_3$ | $\|\omega_{xy}\|_2$ | $-0.005$ | $r_9$ | $\sum_{i=1}^{4}(v_f c_f)$ | $-0.1$ |
| $r_4$ | $\|G_{xy}\|_2$ | $-0.25$ | $r_{10}$ | $\|h-0.35\|_2$ | $-1.0$ |
| $r_5$ | $\|\tau\|_2$ | $-1 \times 10^{-6}$ | | | |

# 3 EXPERIMENTS

We designed 4 experiments to evaluate the performance of the dynamic synapse optimizer in training CPG-network we mentioned above. Because the quadruped robot is not symmetrical in the forward-backward direction, walking forward is more difficult than walking backward, requiring a different Matsuoka CPG structure. Therefore, we use the first structure (Fig.3(a)) to complete the backward walking task, and others (Fig.3(b)(c)) to complete the forward walking task. All experiments share the same reward function and repeated 10 times with different random seeds to ensure statistical reliability.

Additionally, taking the CPG structure in Fig.3(a) as an example, we compare the training effect of DS on our CPG-network with Proximal Policy Optimization (PPO) method. We also compare the robustness of CPG networks and MLP networks under the same parameters in the case of sensor failure. Finally, performance comparison with genetic algorithm (GA) is made. We provide more details and results of the experiment in the appendix, including robustness testing of the hyperparameters of DS.

## 3.1 EFFECTIVENESS AND COMPATIBILITY EXPERIMENT

**Experiment setting.** To evaluate the effectiveness and compatibility of the proposed dynamic synaptic optimizer, we conducted three groups of experiments with varying CPG structures and tasks, while keeping the observation and reward functions identical. Specifically, the first group employed an 8-neuron Matsuoka CPG for the backward walking task, the second group used a 16-neuron Matsuoka CPG for the forward walking task, and the third group utilized an 8-neuron

FitzHugh–Nagumo CPG for the forward walking task. Each experiment was repeated ten times with different random seeds to evaluate statistical reliability.

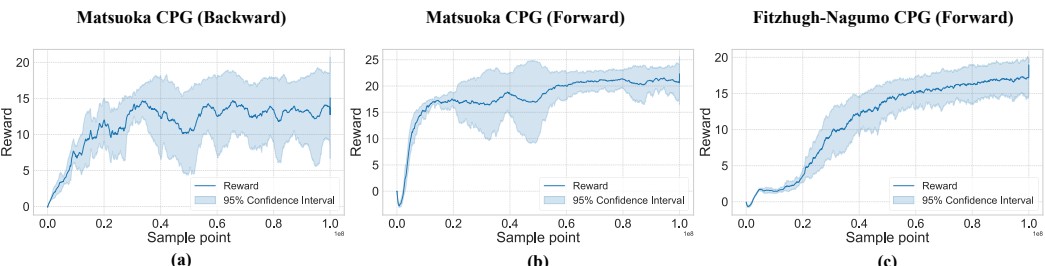

(a)              (b)              (c)

Figure 4: Effectiveness and compatibility experiment. Figure shows the mean reward trajectory across ten replicates and the associated 95% confidence interval of three different CPG structures. (a) The 8-neuron Matsuoka CPG used for backward walking task. (b) The 16-neuron Matsuoka CPG used for forward walking task. (c) The 8-neuron FitzHugh–Nagumo CPG for forward walking task, where each neuron individually generates patterns without lateral connections.

Table 2: Rewards statistics of different CPGs

| Sample Points | 0 | $2.5 \times 10^7$ | $5 \times 10^7$ | $7.5 \times 10^7$ | $1 \times 10^8$ |
|---|---|---|---|---|---|
| 8-MCPG | -0.04±0.03 | 10.49±3.8 | 10.36±6.6 | 12.71±5.16 | 15.08±5.03 |
| 16-MCPG | -0.04±0.01 | 16.86±4.78 | 17.39±8.08 | 20.99±2.4 | 22.33±2.83 |
| 8-FHNCPG | -0.06±0.002 | 6.56±2.72 | 14.37±2.69 | 16.26±2.32 | 18.95±0.61 |

**Results.** For each experiment group, we plotted the average reward curves with 95% confidence intervals over ten independent runs (Fig. 4). Across all tasks, the reward curves exhibit only limited fluctuations and converge steadily to high-reward behaviors, demonstrating the stability of the training process. Furthermore, the ability of the DS optimizer to successfully train controllers based on different CPG models and network structures under a unified observation and reward setting underscores its robustness and broad applicability.

### 3.2 COMPARISON WITH GRADIENT-BASED OPTIMIZER

**Experiment setting.** Training of CPG with DS is compared with CPG trained with PPO under the same conditions. The training results were evaluated by rewards, phase plots of the three joints of the front left (FL) leg, performance in the simulation environment, and performance in the real environment. The task is to train a CPG to control the robot walk backwards at 1 m/s, and the observation and reward functions are the same for the both optimizer. The training is conducted on a machine with Intel I9-13900HX CPU with 32 cores, NVIDIA GeForce RTX 4060 GPU, and 64G memory.

**Result.** The training of two model begin with the same state, so their performance in the simulation environment and the real environment are the same (Fig.5 ①,④), and the period of the CPG can be seen in the phase plot trajectory. When the reward and episode length of DS drop sharply (Fig.5 ②), the robot is prone to fall in the simulation environment, and the phase plot trajectory is messy. In the real environment, since the robot cannot make too large movements, it only performs unstable movements instead of falling directly. However, the CPG seems pass through a bifurcation under the optimization by DS. After the bifurcation, the CPG finds a better pattern, with which the reward increases again. At the end of training, the phase plot trajectory of DS (Fig.5 ③) still maintains a regular "8" shape and reaches a reward value of 20, indicating that DS has successfully trained without destroying the internal structure of the CPG. In contrast, model trained by PPO reaches a lower value of 16 (Fig.5 ⑤) due to gradient exploded, and thus the robot could only move slowly and tremble violently regardless of whether in a simulation environment or a real environment.

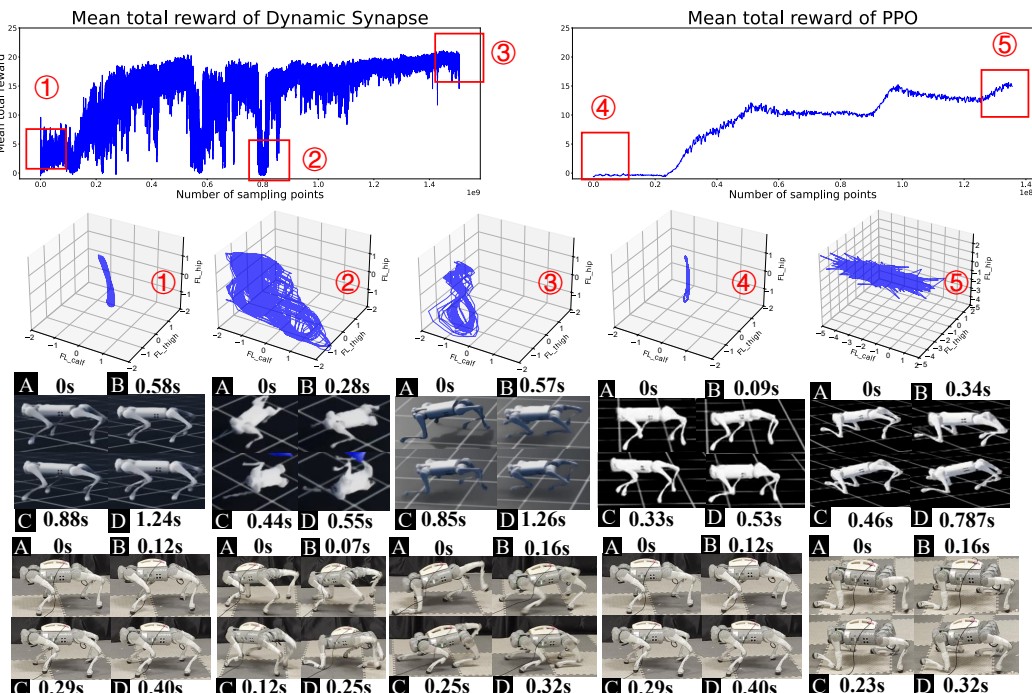

Figure 5: Comparison with gradient-based optimization. The first row shows the mean total reward of the DS and PPO during training. ① is the starting point of DS training, ② is the point where the reward drops sharply during DS training, ③ is the end point of DS training, ④ is the starting point of PPO training, and ⑤ is the end point of PPO training. The second row shows the phase plots between the three joints of the FL leg of the robots. The third row contains screen shots of the robots in the simulation environment. The fourth row contains photos taken on the real robots under control by corresponding models.

### 3.3 ROBUSTNESS EXPERIMENT

**Experiment setting.** In real world application, robots could experience sensor failures, which cause missing observations. Hence, robustness to sensor failures is an important ability of robot controller for real world applications. Animals can adapt to sensor failures. To illustrate a CPG trained with DS optimizer not only has better returns, but also keeps CPGs'advantage with sensory failures, we compare the 8-neuron Matsuoka CPG optimized by DS with an MLP trained by PPO. For a control, the both networks have the same order of magnitude of parameters ($10^3$) when different observations are missing.

To simulate the situation where some observations are missing, we set up five groups of experiments for comparison. The first group has complete observations, the second group sets the projected gravity $G$ to 0, the third group sets the joint positions $q$ to 0, the fourth group sets the last actions $a_{t-1}$ to 0, and the fifth group sets both the joint position $q$ and the last action $a_{t-1}$ to 0.

**Results.** Our model can still roughly walk without projected gravity $G$, joint positions $q$, or the last action $a_{t-1}$ without falling directly, and only the mean total reward and mean episode length decrease. In contrast, the MLP trained with PPO outperforms our model when the observations are complete, but performs poorly when different observations are missing (Tab. 3). More details of the real-world performance of our model can be found in the appendix A.

### 3.4 PERFORMANCE COMPARISON WITH GENETIC ALGORITHM

**Experiment setting.** To further evaluate the effectiveness of the dynamic synaptic optimizer, we employed an 8-neuron Matsuoka CPG for the backward walking task and compared it with a genetic

Table 3: Episode reward and length in robustness test

| Method | Statistics | Complete | $G$ | $q$ | $a_{t-1}$ | $q\&a_{t-1}$ |
|---|---|---|---|---|---|---|
| DS | reward | 20.61 | 19.95 | 16.15 | 11.03 | 8.02 |
|  | length | 981.5 | 942.92 | 944.55 | 990.44 | 889.44 |
| PPO | reward | 41 | 1.95 | -0.43 | -0.25 | -0.14 |
|  | length | 1000 | 90.51 | 64.41 | 18.66 | 19.91 |

algorithm (GA) under identical conditions (same CPG structure, observation and reward functions, and environment setup). The GA used a population size of 100, mutation probability of 0.02, and was run for up to 200 generations.

Table 4: Episode rewards of GA along generations

| Generations | 10 | 20 | 30 | 40 | 50 | 60 | 64 |
|---|---|---|---|---|---|---|---|
| reward | -2.526 | -17.79 | 0.3117 | -0.3549 | -69.95 | 0.2424 | -inf |

**Results.** We observed that the genetic algorithm (GA) often collapsed very early, sometimes as soon as the fifth generation. Despite extensive hyperparameter tuning, we were only able to delay this collapse to roughly the 60th generation on average. This instability caused the robot to fall frequently or produced NaN states, terminating training prematurely. In contrast, our proposed dynamic synaptic optimizer maintained training stability throughout and consistently converged to high-reward, robust gaits without early collapse.

These results suggest that gradient-free search struggles to discover viable parameter sets for the CPG in this high-dimensional continuous control setting, whereas our optimizer can efficiently guide learning toward stable locomotion.

## 4 DISCUSSIONS

Traditional reinforcement learning algorithms (e.g., PPO) have excellent performance in training ANN networks, but there are problems, such as long iteration time and poor training effect arising from gradient vanishment/explosion when training networks contain recursive structure, such as CPG. Our proposed optimizer utilizes a learning mechanism based on weight self-fluctuation and feedback modulation, which can well avoid the above problems. DS allows online training of CPGs combined with a general network, which provides a series of new possible RNN-based architectures for robot motion control.

Combining the advantages of biological plausible reinforcement learning paradigm and stable rhythmic generation from inherent properties of CPG as a framework, it can optimize robot motion control while not overly relying on observations. This framework provides new ideas for research of reinforcement learning.

This work is limited in that, though the DS contains a parallel computing mechanism, it does not fully utilize the parallel simulation environment, resulting in no significant reduction in training time. We are working on a further version of the optimizer, and the preliminary experiments suggest an order of speed improvement.

There is still a gap between the performance of the CPG-network trained by DS and the traditional MLP network trained by PPO. But it must be pointed out that our work focuses on whether the method can train an RNN or biologically plausible neural circuit models to control a robot for real-world locomotion, and we confirmed this by applying it on CPG. In the next step, we will further explore more general applications across diverse tasks and terrains.

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

## A APPENDIX

EXPONENTIAL APPROXIMATION OF DS LEARNING RULE

The equations of the learning rule are formulated as differential equations (Eq.3). To simplify computation, we actually use the Euler method for discrete iteration. To prevent the amplitude from becoming negative in the training epochs while minimizing conditional checks, we modify the details of amplitude modulation by implementing an exponential approximation mechanism:

$$A_{t+1} = A_t e^{-\beta R_t \Delta t} \tag{7}$$

where the subscript $t$ denotes the moment t, e the natural logarithm base and $\Delta t$ the iteration step size. Typically, by controlling the overall learning rate(not the $\alpha$ above), convergence rate $\beta$, and size of iteration step $\Delta t$, the incremental of $A$ at each step remains minimal (below $1 \times 10^{-4}$) so that the difference between $A_{t+1}$ and $A_t$ derived from exponential approximation is almost the same as that computed via the Euler method due to infinitesimal equivalence. However, it is acknowledged that under conditions of large absolute reward values, this mechanism may induce convergence suppression and exploration enhancement. We argue that this is not necessarily a bad thing for network optimization: the promotion of exploration from negative reward facilitates the search of weight space, while the small suppression of amplitude convergence at high reward may ultimately lead to higher robustness.

SYMBOLS OF THE QUADRUPED ROBOT

Full description of symbols of the quadruped robot are listed here.

Table 5: Symbols of quadruped robot

| Symbol | Description |
| --- | --- |
| $a$ | Action |
| $v_{xy}$ | Linear velocity along $X/Y$ axes |
| $v_z$ | Linear velocity along $Z$ axis |
| $\omega_{xy}$ | Angular velocity along $X/Y$ axes |
| $\omega_z$ | Angular velocity along $Z$ axis |
| $q, \dot{q}, \ddot{q}$ | Joint positions/velocities/accelerations |
| $\tau$ | Joint torques |
| $G$ | Normalized projected gravity |
| $G_{xy}$ | Projected gravity on $X/Y$ axes |
| $c_f$ | Foot contact statuses |
| $v_f$ | Foot velocity on $X/Y$ axes |
| $v_c$ | Target linear velocity |
| $\omega_c$ | Target angular velocity |
| $h$ | Robot trunk height |

PERFORMANCE OF REAL ROBOT IN ROBUSTNESS EXPERIMENT

Gait performance of the real robot under CPG control trained by DS is shown in the following figure:

**Full observation**

| 0s | 0.16s | 0.25s | 0.32s | 0.42s | 0.50s | 0.62s | 0.70s |

**Missing projected gravity**

| 0s | 0.03s | 0.20s | 0.30s | 0.55s | 0.58s | 0.77s | 0.86s |

**Missing joint positions**

| 0s | 0.09s | 0.25s | 0.42s | 0.63s | 0.74s | 0.92s | 1.04s |

**Missing last action**

| 0s | 0.06s | 0.25s | 0.35s | 0.47s | 0.56s | 0.76s | 0.86s |

**Missing joint positions and last action**

| 0s | 0.08s | 0.27s | 0.39s | 0.52s | 0.88s | 1.14s | 1.24s |

Figure 6: Robustness test experiment. When projected gravity is missing, our model's walking ability is barely reduced. When joint positions are missing, our model's hind legs have difficulty extending backwards and slipping, which affects walking. When the last actions are missing, our model's movements start to slow down, and its walking speed decreases. When joint positions and last actions are missing simultaneously, our model not only walks slowly but also has difficulty keeping a straight line, but still does not fall.

THE ROBUSTNESS OF DYNAMIC SYNAPSE HYPERPARAMETERS

We conduct ablation studies on six hyperparameters by comparing smaller/larger values against the original values used in our experiments. The results are analyzed as follows.

**amp_init (Amplitude Initialization)**    This parameter affects the instantaneous weight of initialization as well as the early amplitude. Larger values enhance exploration, but this can lead to a more drastic decay of early rewards; Smaller values lead to a more stable uptrend, but there is a chance of getting stuck in the local optimal solution in the later stages of training. According to our experimental results, generally speaking, as long as it is not a too large or too small value, it will not have a significant impact on the final reward.

Table 6: amp_init

| Sample Points | $1.25 \times 10^7$ | $2.5 \times 10^7$ | $3.75 \times 10^7$ | $5 \times 10^7$ | $6.25 \times 10^7$ | $7.5 \times 10^7$ | $8.75 \times 10^7$ | $1 \times 10^8$ |
|---|---|---|---|---|---|---|---|---|
| smaller | 16.0 | 19.14 | 20.04 | 18.65 | 21.53 | 20.37 | 22.36 | 22.88 |
| regular | 16.34 | 18.2 | 18.89 | 20.3 | 20.35 | 20.95 | 21.33 | 22.11 |
| bigger | 7.56 | 16.51 | 19.12 | 9.86 | 18.55 | 18.28 | 18.15 | 18.95 |

**amp_update_rate (Amplitude Update Rate)**    This parameter affects the changing rate of amplitude, which in turn affects the convergence of dynamic synapses. Larger values lead to premature convergence and failure to discover the optimum; Smaller values may enhance the robustness of the model, but risk unstable late-stage training, which will lead to convergence difficulties.

Table 7: amp_update

| Sample Points | $1.25 \times 10^7$ | $2.5 \times 10^7$ | $3.75 \times 10^7$ | $5 \times 10^7$ | $6.25 \times 10^7$ | $7.5 \times 10^7$ | $8.75 \times 10^7$ | $1 \times 10^8$ |
|---|---|---|---|---|---|---|---|---|
| smaller | 15.77 | 18.62 | 19.94 | 20.08 | 21.25 | 21.05 | 20.68 | 21.95 |
| regular | 16.34 | 18.2 | 18.89 | 20.3 | 20.35 | 20.95 | 21.33 | 22.11 |
| bigger | 15.56 | 18.31 | 18.48 | 18.42 | 18.45 | 18.55 | 18.45 | 18.46 |

**lr (Learning Rate)**   This parameter affects the changing rate of the weight center and amplitude. It is a parameter that regulates the overall rate of change. Therefore, too large results in early convergence or unstable learning, or too small results in prolonged training time. However, in general, if the other parameters are set properly, slight changes (we tried to enlarge it by twice or reduce it by half) will not affect the final result.

Table 8: lr

| Sample Points | $1.25 \times 10^7$ | $2.5 \times 10^7$ | $3.75 \times 10^7$ | $5 \times 10^7$ | $6.25 \times 10^7$ | $7.5 \times 10^7$ | $8.75 \times 10^7$ | $1 \times 10^8$ |
|---|---|---|---|---|---|---|---|---|
| smaller | 13.29 | 18.84 | 20.01 | 20.52 | 20.86 | 21.03 | 21.15 | 21.58 |
| regular | 16.34 | 18.2 | 18.89 | 20.3 | 20.35 | 20.95 | 21.33 | 22.11 |
| bigger | 17.09 | 17.64 | 17.93 | 20.08 | 20.27 | 20.67 | 21.16 | 21.54 |

**period (Oscillation Period)**   This parameter controls the mean fluctuation period of dynamic synapses, and this period should match the causality period between action and reward. A period that is too short results in that when a reward approaches, the current parameter is away from the position that causes the reward; a period that is too long causes prolonged training time. This parameter should be estimated according to the tasks (but this is usually not difficult). For example, for periodic tasks, it is recommended that a fluctuation period should allow the robot to complete 2 - 4 times of behavior.

Table 9: period

| Sample Points | $1.25 \times 10^7$ | $2.5 \times 10^7$ | $3.75 \times 10^7$ | $5 \times 10^7$ | $6.25 \times 10^7$ | $7.5 \times 10^7$ | $8.75 \times 10^7$ | $1 \times 10^8$ |
|---|---|---|---|---|---|---|---|---|
| smaller | 0.3 | 0.82 | 0.46 | 0.35 | 0.31 | 0.37 | 0.34 | 0.29 |
| regular | 16.34 | 18.2 | 18.89 | 20.3 | 20.35 | 20.95 | 21.33 | 22.11 |
| bigger | 16.92 | 17.78 | 18.55 | 19.24 | 19.7 | 20.27 | 19.92 | 20.42 |

**period_var (Oscillation Period Variance)**   This parameter controls the variance of the fluctuation period of synapses' weight to make the oscillation free from phase-lock. It usually should be small, for example, 5% of the period. An extremely small value will lead to almost constant periods and phase-lock, limit exploration diversity, and thus increase the probability of falling into a local optimum.

Table 10: period_var

| Sample Points | $1.25 \times 10^7$ | $2.5 \times 10^7$ | $3.75 \times 10^7$ | $5 \times 10^7$ | $6.25 \times 10^7$ | $7.5 \times 10^7$ | $8.75 \times 10^7$ | $1 \times 10^8$ |
|---|---|---|---|---|---|---|---|---|
| smaller | 0.42 | 1.51 | 3.34 | 2.83 | 0.96 | 5.01 | 1.51 | 3.18 |
| regular | 16.34 | 18.2 | 18.89 | 20.3 | 20.35 | 20.95 | 21.33 | 22.11 |
| bigger | 9.84 | 17.2 | 17.18 | 18.25 | 18.35 | 3.7 | 18.28 | |

**weight_centre_update_rate (Weight Center Update Rate)**   This parameter controls the rate of change of the center of the fluctuation in dynamic synapses. Large values help the model quickly find a better state but reduce the stability of exploration and learning. Small values result in prolonged training time. In fact, this parameter is more similar to learning_rate in training general artificial neural networks.

Table 11: weight_center_update_rate

| Sample Points | $1.25 \times 10^7$ | $2.5 \times 10^7$ | $3.75 \times 10^7$ | $5 \times 10^7$ | $6.25 \times 10^7$ | $7.5 \times 10^7$ | $8.75 \times 10^7$ | $1 \times 10^8$ |
|---|---|---|---|---|---|---|---|---|
| smaller | 15.72 | 18.69 | 18.73 | 19.63 | 19.96 | 20.17 | 20.37 | 20.5 |
| regular | 16.34 | 18.2 | 18.89 | 20.3 | 20.35 | 20.95 | 21.33 | 22.11 |
| bigger | 3.48 | 4.07 | 0.02 | 9.01 | 10.35 | 13.98 | | |

