# OpenReview forum: "A Bio-inspired Gradient-Free Learning Framework for CPG-based Neural Networks in Robot Locomotion"
_ICLR.cc/2026/Conference — Submitted to ICLR 2026_

### Official Review · Reviewer_EeAs · 2025-11-01

**Soundness:** 3
**Presentation:** 3
**Contribution:** 2
**Rating:** 2
**Confidence:** 4

**Summary:**

This paper introduces a Dynamic Synapse (DS) learning rule, a gradient-free, biologically inspired optimization approach in which each network weight oscillates sinusoidally, and its center is updated based on received rewards. The authors use DS to train central pattern generator controllers for quadruped locomotion, both in simulation and on a real robot. They claim that DS can train architectures where PPO fails due to gradient instability and that DS-trained controllers exhibit robustness to sensor failures.

**Strengths:**

- Novel yet simple gradient-free mechanism derived from neurodynamics, implemented efficiently in PyTorch for parallel rollouts.
- Demonstrates stable training of CPG-RNN controllers that PPO struggles with.
- Includes real-robot results and robustness evaluations, which strengthen the empirical credibility.
- Provides an interesting connection between biologically inspired oscillatory dynamics and practical control learning.

**Weaknesses:**

1. The approach should be compared against stronger evolutionary algorithms (CMA-ES) and well-tuned PPO/SAC.
2. Provide ablations: (a) original vs. new DS formulation, (b) which parameters are DS-trained, (c) reward sensitivity.
3. The approach should also be tested on multi-skill or sparse-reward tasks to assess generality.
4. Clarify total learnable parameter count.
5. Investigate using one shared CPG topology conditioned on commands, rather than per-task networks.

**Questions:**

- “Therefore, we use the first structure (Fig.3(a)) to complete the backward
walking task, and others (Fig.3(b)(c)) to complete the forward walking task. “ -> seems complicated for just a walking task. How would this scale to other tasks? Would this suggest you need a new network for each task?
- “We observed that the genetic algorithm (GA) often collapsed very early, sometimes as
soon as the fifth generation” -> I'm surprised by this. I would suggest trying a much larger population size
-  “his instability caused the robot to fall frequently or produced NaN states, terminating training prematurely” -> For me, this rather points to some kind of bug. There shouldn't be any NaN numbers because training converges.

---

> ### Author Response · Authors · 2025-11-23
>
> Thank you for your insightful comments and constructive suggestions, which help to improve the paper. And thank you for pointing out the strength: gradient-free mechanism derived from neurodynamics, efficient implementation of Pytorch, stable training of CPG-RNN controller, real-robots results, robustness evaluations, and the connection between bio-inspired oscillatory and practical learning.
>
> Below is our response to your review comments.
> ## Main focus of our work(for Weakness 3&5, Questions 1)
>
> We would kindly highlight that the focus of this paper is to **explore the boundary of the DS in optimising CPG-based RNNs for a physical robot**, which is difficult for gradient-based optimisation methods, instead of competing with conventional deep reinforcement learning methods in robot control.
>
> ## Regarding the comparison with Genetic Algorithm and PPO(for Weakness 1, Question 2)
> We fully understand your surprise at the poor performance of the evolutionary algorithm in our experiment, and the natural inclination to suggest employing a stronger evolutionary algorithm and a well-tuned PPO. However, their primary shortcomings are structural in nature.
>
> The major drawback of the evolutionary algorithm is that the randomness of crossover and mutation results in a sudden change of the robot's action, which is hazardous when it is applied to a real robot, while the exploration of our method, dynamic synapse, is continuous and soft, allowing our algorithm to perform actual training on physical robots. And PPO is prone to exploding gradients when training essentially recurrent neural networks such as CPG, which possess complex temporal dependencies and recurrent connections, without special processing (such as clipping or gating mechanisms). Therefore, these two methods; applicability in this scenario is somewhat limited, further highlighting the advantages and stability of our proposed weight perturbation-based method in traning such structures for real-world locomotion.
>
> ## Regarding learnable parameters(for Weakness 2(b)&4)
> The point you raised is indeed crucial, and it was an oversight on our part not to clarify which parameters were optimized. We will add this information to the paper before it is finalized.
>
> As we mentioned in the paper, our network contains three layers: Obs-layer, CPG-layer, and Act-layer. For an $i$-neuro CPG network, Obs-layer has $i\times 45$ weights and $i$ biases, CPG-layer has $i\times 12$ weights and $12$ biases, and Act-layer has $i^2$ weights and $i$ biases. Hence, there are $i^2+47i+12$ parameters in an $i$-neuro CPG network to be optimized. Note that in our work, $i$ has only two values, $8$ and $16$.
>
> ## Regarding reward sensitivity and sparseness(for Weakness 2(c) and 3)
>
> At the current stage, the reward for DS should be dense, as it is only a synapse-level reinforcement learning rule, which does not predict the potential reward of the future. In the animal brain, there are reward circuits converting sparse rewards for synapse-level learning, such as the circuit in the mushroom body. We are investigating it for more robust reward acceptance. For the current stage, if the original reward cannot function well, we use an adaptation mechanism to shift the mean reward to zero, which helps the training.
>
> ## Regarding for the NaN numbers(for Question 3)
>
> Methods like GA can cause unstable dynamics in an RNN, such as CPG. A stable CPG relies on its stable limit cycle; the states in its domain of attraction fall to the limit cycle for a repetitive action. However, the mutation and cross operation with GA can break the system, changing the stable limit cycle into an unstable system, which pushes states away to infinity, which is then likely to cause 0 and NaN in later computation. As DS explores within a limited region for each step, it is numerically more stable.
>
> ## Investivate Different CPGs instead of one CPG (for Question 5)
>
> Thank you for the suggestion. Controlling a robot conditioned on commands is a commen way for robot reinforcement learning, and we do it too for non-CPG based models, and we will investigate it in the next stage for CPG based models. We have not presented it in this paper because this paper focuses on explorating the boundary of the DS instead of the application of RL, so we tested it on different models instead of only one. As DS does trained these CPG-based networks, we proved that DS can train various models for a physical robot.
>
> ## Ablation studies of hyperparameters(for Weakness 2(a))
> Please see the further comment.
>
> Thank you again for your insightful comments and suggestions. We kindly invite you to evaluate the work based on the clarified focus outlined in our response, and we look forward to your further feedback.

---

> > ### Author Response · Authors · 2025-11-23
> >
> > ## Ablation studies of hyperparameters(for Weakness 2(a))
> > We conduct ablation studies on six hyperparameters by comparing smaller/larger values against the original values used in our experiments. The results are analyzed as follows:
> >
> > 1. amp_init（Amplitude Initialization）
> > This parameter affects the instantaneous weight of initialization as well as the early amplitude.
> > Larger values enhance exploration, but this can lead to a more drastic decay of early rewards.
> > Smaller values lead to a more stable uptrend, but there is a chance of getting stuck in the local optimal solution in the later stages of training.
> > According to our experimental results, generally speaking, as long as it is not a too large or too small value, it will not have a significant impact on the final reward.
> >
> >     | Sample point | 0 | 1.25e7 | 2.5e7 | 3.75e7 | 5e7 | 6.25e7 | 7.5e7 | 8.75e7 | 1e8 |
> >     | ---: | ---: | ---: | ---: | ---: | ---: | ---: | ---: | ---: | ---: |
> >     | smaller |  | 16.0 | 19.14 | 20.04 | 18.65 | 21.53 | 20.37 | 22.36 | 22.88 |
> >     | regular |  | 16.34 | 18.2 | 18.89 | 20.3 | 20.35 | 20.95 | 21.33 | 22.11 |
> >     | bigger | -0.06 | 7.56 | 16.51 | 19.12 | 9.86 | 18.55 | 18.28 | 18.15 | 18.95 |
> >
> > 2. amp_update_rate (Amplitude Update Rate)
> > This parameter affects the changing rate of amplitude, which in turn affects the convergence of dynamic synapses.
> > Larger values lead to premature convergence and failure to discover the global optimum;
> > Smaller values may enhance the robustness of the model, but risk unstable late-stage training, which will lead to convergence difficulties.
> >
> >     | Sample point | 0 | 1.25e7 | 2.5e7 | 3.75e7 | 5e7 | 6.25e7 | 7.5e7 | 8.75e7 | 1e8 |
> >     | ---: | ---: | ---: | ---: | ---: | ---: | ---: | ---: | ---: | ---: |
> >     | smaller |  | 15.77 | 18.62 | 19.94 | 20.08 | 21.25 | 21.05 | 20.68 | 21.95 |
> >     | regular |  | 16.34 | 18.2 | 18.89 | 20.3 | 20.35 | 20.95 | 21.33 | 22.11 |
> >     | bigger |  | 15.56 | 18.31 | 18.48 | 18.42 | 18.45 | 18.55 | 18.45 | 18.46 |
> >
> > 3. lr (Learning Rate)
> > This parameter affects the changing rate of the weight center and amplitude. It is a parameter that regulates the overall rate of change. Therefore, too large or too small a parameter may have the same effect as the above two parameters. However, in general, if the other parameters are set properly, slight changes (we tried to enlarge it by twice or reduce it by half) will not affect the final result.
> >
> >     | Sample point | 0 | 1.25e7 | 2.5e7 | 3.75e7 | 5e7 | 6.25e7 | 7.5e7 | 8.75e7 | 1e8 |
> >     | ---: | ---: | ---: | ---: | ---: | ---: | ---: | ---: | ---: | ---: |
> >     | smaller |  | 13.29 | 18.84 | 20.01 | 20.52 | 20.86 | 21.03 | 21.15 | 21.58 |
> >     | regular |  | 16.34 | 18.2 | 18.89 | 20.3 | 20.35 | 20.95 | 21.33 | 22.11 |
> >     | bigger |  | 17.09 | 17.64 | 17.93 | 20.08 | 20.27 | 20.67 | 21.16 | 21.54 |
> >
> > 4. period (Oscillation Period)
> > This parameter controls the normal mean used to generate the fluctuation period of dynamic synapses, which in turn affects the exploration of parameter space and whether the synapses have "effective" learning. In other words, it is probably the most important of these six parameters.
> > If the period is too small, that is, the dynamic synapses fluctuate rapidly, then when the synapses obtain the reward caused by the current action, the weights may have moved away from the state that caused the action due to fluctuation, and then the changes of the fluctuation centers will not match the changes of the instantaneous weights, resulting in wrong learning.
> > If the period is too large, the too slow fluctuations may lead to a local optimum. Take the walking task as an example, if one fluctuating period can allow the robot to complete dozens of behavioral steps, then the slight changes of weights in one behavioral step will result in limited exploration and thus risking falling into a local optimum.
> > Therefore, we need to point out that this parameter should be estimated according to the tasks (but this is usually not difficult). For example, for periodic tasks, it is recommended that a fluctuation period should allow the robot to complete 2 - 4 times of the behavior. Sincerely thanks again for your reminder, we will supplement this section to the Discussion section of the paper.
> >
> >     | Sample point | 0 | 1.25e7 | 2.5e7 | 3.75e7 | 5e7 | 6.25e7 | 7.5e7 | 8.75e7 | 1e8 |
> >     | ---: | ---: | ---: | ---: | ---: | ---: | ---: | ---: | ---: | ---: |
> >     | smaller |  | 0.3 | 0.82 | 0.46 | 0.35 | 0.31 | 0.37 | 0.34 | 0.29 |
> >     | regular |  | 16.34 | 18.2 | 18.89 | 20.3 | 20.35 | 20.95 | 21.33 | 22.11 |
> >     | bigger | -0.06 | 16.92 | 17.78 | 18.55 | 19.24 | 19.7 | 20.27 | 19.92 | 20.42 |

---

> > > ### Author Response · Authors · 2025-11-23
> > >
> > > 5. period_var (Oscillation Period Variance)
> > > This parameter controls the normal variance used to generate the fluctuation period of dynamic synapses.
> > > Large values will lead to some too large/small periods generated during the exploration process, which may have the same consequences explained above in the parameter period.
> > > Small values will lead to almost constant periods, limit exploration diversity, and thus increase the probability of falling into a local optimum.
> > >
> > >     | Sample point | 0 | 1.25e7 | 2.5e7 | 3.75e7 | 5e7 | 6.25e7 | 7.5e7 | 8.75e7 | 1e8 |
> > >     | ---: | ---: | ---: | ---: | ---: | ---: | ---: | ---: | ---: | ---: |
> > >     | smaller |  | 0.42 | 1.51 | 3.34 | 2.83 | 0.96 | 5.01 | 1.51 | 3.18 |
> > >     | regular |  | 16.34 | 18.2 | 18.89 | 20.3 | 20.35 | 20.95 | 21.33 | 22.11 |
> > >     | bigger | -0.03 | 9.84 | 17.2 | 17.18 | 18.25 | 18.35 | 3.7 | 18.28 |  ||
> > >
> > > 6. weight_centre_update_rate (Weight Center Update Rate)
> > > This parameter controls the changing rate of the center of the fluctuation in dynamic synapses.
> > > Large values help the model quickly find a state better than the initial one in the early stage, but in the middle and late stages of training, it will lead to more drastic mutations of the actions, reduce the stability of the robot during exploration, and even lead to the inability to converge and training collapse.
> > > Small values, on the contrary, will lead to inefficient training or fall into the local optimal value and converge prematurely. In fact, this parameter is more similar to learning_rate in training general artificial neural networks.
> > >
> > >     | Sample point | 0 | 1.25e7 | 2.5e7 | 3.75e7 | 5e7 | 6.25e7 | 7.5e7 | 8.75e7 | 1e8 |
> > >     | ---: | ---: | ---: | ---: | ---: | ---: | ---: | ---: | ---: | ---: |
> > >     | smaller |  | 15.72 | 18.69 | 18.73 | 19.63 | 19.96 | 20.17 | 20.37 | 20.5 |
> > >     | regular |  | 16.34 | 18.2 | 18.89 | 20.3 | 20.35 | 20.95 | 21.33 | 22.11 |
> > >     | bigger |  | 3.48 | 4.07 | 0.02 | 9.01 | 10.35 | 13.98 |  |  |

---

> > ### Comment · Reviewer_EeAs · 2025-11-28
> >
> > Thank you for the clarifications. Some more additional points.
> >
> > “And PPO is prone to exploding gradients when training essentially recurrent neural networks such as CPG,”-> In this case it would be good to do an experiment where you train with PPO on an architecture with and without CPG to show this issue does actually exist.
> >
> > Also what about using something like CMA-ES?
> >
> > In regards to the NaN issue. Did you try to clip the weights after the mutations? That should alleviate this issue.

---

### Official Review · Reviewer_atth · 2025-11-01

**Soundness:** 3
**Presentation:** 3
**Contribution:** 2
**Rating:** 4
**Confidence:** 2

**Summary:**

This paper presents a bio-inspired learning framework that enables effective training of recurrent neural networks with Central Pattern Generators (CPGs) for robot reinforcement learning. The proposed method achieves robust quadrupedal locomotion and improved resilience to sensor failure.

**Strengths:**

The paper is well written and has hardware demonstrations. The design choices are nicely ablated in both simulation and hardware. I also like the idea of integrating CPGs into RL (although I am unsure about its benefits as discussed in weaknesses below).

**Weaknesses:**

1. The paradigm in legged locomotion seems to use fast and efficient simulators, e.g., Mujoco playground (https://arxiv.org/abs/2502.08844) to train policies efficiently in simulation and transfer to real. I am not quite sure how the proposed approach would be better than the more common strategy and what its underlying benefits are.

2. In addition, the proposed method does not benefit from fast parallel simulators, which are widely available and highly efficient.

3. The hardware demonstrations and the simulation tasks considered in this work are not at all dynamic compared to the SOTA methods for legged locomotion (https://arxiv.org/pdf/2010.11251, https://ieeexplore.ieee.org/abstract/document/10225268)

**Questions:**

See weaknesses

---

> ### Author Response · Authors · 2025-11-23
>
> Thank you very much for your insightful comments, and we appreciate your recognition of our work, which contains hardware demonstrations, ablation studies, and integrates CPGs into RL.
>
> Below is our response to your questions.
>
> ## Motivation and benefits (for Weakness 1)
>
> Training the robot in simulation is a common approach in reinforcement learning, as these methods are usually data-inefficient and require resetting the robot frequently to actively collect data while the robot has not yet fallen. However, real animals do not learn in this way. They learn within a few trials and do not require resets. There is still a long way to go before we can reproduce these abilities by mimicking neural mechanisms. Dynamic synapses are a potential way to make this easier, as they do not require gradients; thus, biologically plausible neural circuits can be more straightforwardly applied to robot learning. The CPG-based networks in this paper are only one of the first steps toward more bio-inspired neural networks with great potential. With the work presented in this paper, we confirmed that DS is able to train CPG-based RNNs for a physical robot, not only in simple simulated control tasks, and that the robustness of CPGs to sensor failure is preserved after training. We will investigate more complex circuits heading toward more bio-inspired networks for robot learning.
>
> ## Main focus and benefits of our work (for Weaknesses 1 & 3)
>
> We acknowledged that there is still a performance gap between the CPG-network trained by DS and more common strategies (for example, a traditional MLP network trained by PPO) at the end of the paper’s discussion section. However, **our work focuses on** whether the DS method can train a robot for real-world locomotion, and we confirmed this by applying it to a CPG. In the next step, we will try more dynamic tasks. This work does not intend to compete with conventional deep reinforcement learning methods in robot control, but we show that a CPG-based RNN trained with our method is more robust to sensor failure than a gradient-based method.
>
> The locomotion task is not as dynamic as in SOTA methods, which do not follow biological constraints for biological compatibility. But as discussed above, we believe that with DS simplifying the optimization of biologically plausible circuits, we are moving toward more efficient and natural learning behavior for real-world learning.
>
> ## Insufficient utilization of the parallel simulation environment (for Weakness 2)
>
> Our method currently uses the same spontaneous weights for all simulated instances, and different instance returns different rewards, which accelerates the training.  “Does not fully utilize the parallel simulation environment” in the discussion of the paper means that our method had not realized a full parallelization (due to time constraints) by sharing the weight center but independent instantaneous weights, so that different policy instances could use different instance weights. This would achieve full simulation-level parallelization and greatly improve training efficiency.
>
> But we have now successfully adopted it, and we are happy to add the relevant experimental results to the paper should they be necessary for the paper’s acceptance.
>
> Thank you again for your insightful comments and suggestions. Hope the above response addresses your concerns, and we look forward to your further feedback.

---

### Official Review · Reviewer_zA2W · 2025-11-03

**Soundness:** 2
**Presentation:** 2
**Contribution:** 1
**Rating:** 2
**Confidence:** 5

**Summary:**

This paper proposes a bio-inspired gradient-free learning framework to address the issue that Central Pattern Generators (CPGs) – which are effective for robot locomotion control but rarely used in reinforcement learning (RL) due to gradient vanishing or explosion caused by their inherent recursive connections – are difficult to optimize via gradient-based methods. The core of this framework is a Dynamic Synapse (DS) optimizer, which realizes parameter exploration through weight fluctuations and convergence through reward feedback, and it is used to train a three-layer network (including Obs-layer, CPG layer with specific oscillators, and Act-layer) for quadrupedal locomotion on the Unitree go1 robot. Experiments show that the framework can successfully train the CPG-based network, whereas Proximal Policy Optimization (PPO) fails due to gradient problems; the CPG-based network is more robust to sensor failures than PPO-trained feedforward networks, and it avoids the early training collapse of Genetic Algorithms (GA). Although the DS optimizer supports parallel training, it does not fully utilize parallel simulation environments, and its performance still lags behind PPO-trained MLP networks, but the work overall lowers the barrier to applying recurrent neural networks in robot RL.

**Strengths:**

1. The paper proposes a bio-inspired gradient-free Dynamic Synapse (DS) optimizer that effectively addresses the gradient vanishing/explosion issue caused by the inherent recursive connections of Central Pattern Generators (CPGs), enabling successful training of CPG-based neural networks for quadruped robot locomotion— a challenge that gradient-based methods like Proximal Policy Optimization (PPO) fail to overcome , .
2. The CPG-based network trained with the proposed framework demonstrates stronger robustness to sensor failures compared to PPO-trained feedforward networks, which is a critical advantage for real-world robot applications where sensor malfunctions may occur.

**Weaknesses:**

1. The Dynamic Synapse (DS) optimizer, despite supporting parallel training through PyTorch implementation, fails to fully leverage parallel simulation environments, resulting in no significant reduction in training time, which limits its efficiency for large-scale or time-sensitive robot locomotion training tasks .
2. There remains a performance gap between the CPG-network trained by the DS optimizer and the traditional Multilayer Perception (MLP) network trained by Proximal Policy Optimization (PPO), indicating the proposed framework still lags behind existing mainstream methods in terms of locomotion control performance under complete observation conditions .
3. The framework’s validation is relatively limited to basic quadruped locomotion tasks (forward and backward walking) on the Unitree go1 robot, without extending to more complex scenarios such as locomotion over challenging terrains or multi-skill motion control, restricting the demonstration of its general applicability .
4. The DS optimizer’s hyperparameters (e.g., amplitude initialization, oscillation period) require careful tuning based on specific tasks (e.g., matching the oscillation period to the causality period between action and reward), and improper settings may lead to issues like early convergence, local optima, or prolonged training time, increasing the complexity of practical application , .

**Questions:**

1.  Reward hacking is a common issue in reinforcement learning. Have you observed any cases where the robot achieves high rewards through exploiting loopholes in the designed reward function (e.g., abnormal locomotion without effective movement)? If not, what evidence supports that the reward function can avoid such problems?
2.  The DS optimizer relies heavily on reward signals for parameter adjustment. How does noisy or inaccurate reward (e.g., from sensor errors) affect its convergence and stability? Do you have experimental evaluations on the optimizer’s robustness to reward noise?
3.  The DS-CPG network lags behind PPO-MLP in performance under complete observations. Could this gap be narrowed by adjusting the reward function’s weight distribution or adding new terms? Have you conducted relevant tests?
4.  The reward terms for locomotion metrics (e.g., trunk height, foot sliding) – are they designed based on prior knowledge or systematic optimization (e.g., hyperparameter search)? Would this design limit adaptability to other legged robots (e.g., hexapods)?
5.  You compared DS with GA and PPO in stability, but not in reward efficiency (steps to reach a target reward). Does DS require more training steps to converge than GA when both are stable? How does it compare to PPO (excluding gradient issues)?
6.  The DS optimizer’s design is tailored to periodic locomotion tasks. For non-periodic tasks (e.g., robotic manipulation), what key adjustments (e.g., fluctuation period, learning rules) would be needed? Is there preliminary evidence for its applicability?
7.  You mentioned underutilization of parallel simulation. What specific bottlenecks exist in the current parallel implementation? Do you have plans to optimize it (e.g., adjusting parameter sharing mechanisms) to reduce training time?

---

> ### Author Response · Authors · 2025-11-25
>
> Thank you very much for your comments. We appreciate that you point out our bio-inspired gradient-free DS optimizer effectively addresses the gradient vanishing/explosion issue and successfully trained a CPG-based neural network for quadruped robot locomotion, and notice that CPG demonstrates stronger robustness to sensor failures compared to PPO.
>
> Below is our response to your review comments.
>
> ## About the parallelization (for Weakness 1, Question 7)
> Our method currently uses the same spontaneous weights for all simulated instances, and different instance returns different rewards, which accelerates the training. “Does not fully utilize the parallel simulation environment” in the discussion of the paper means that our method had not realized a full parallelization (due to time constraints) by sharing the weight center but independent instantaneous weights, so that different policy instances could use different instance weights. This would achieve full simulation-level parallelization and greatly improve training efficiency.
>
> But now we successfully make full use of the parallel simulation environment. We have adjusted our parameter sharing mechanism such that different policy instances could use different instance-specific weights while sharing the weight fluctuation center, and the efficiency of training has been improved. And we are happy to add the relevant experimental results to the paper, should they be necessary for the paper’s acceptance.
>
> ## Focus of our work (for Weakness 2 & 3, Question 3)
> Real animals can learn within a few trials and do not require resets, which is impossible with the current RL methods. There is still a long way to go before we can reproduce these abilities by mimicking neural mechanisms. Dynamic synapses are a potential way to make this easier, as they do not require gradients; thus, biologically plausible neural circuits can be more straightforwardly applied to robot learning. The CPG-based networks in this paper are only one of the first steps toward more bio-inspired neural networks with great potential. With the work presented in this paper, we confirmed that DS is able to train CPG-based RNNs for a physical robot, not only in simple simulated control tasks, and that the robustness of CPGs to sensor failure is preserved after training. We will investigate more complex circuits heading toward more bio-inspired networks for robot learning.
>
> Hence, our work focuses on evaluating the potential of DS for introducing more biologically inspired networks by using it to train a robot for real-world locomotion, instead of benchmarking for specific tasks to outperform SOTA methods, although it does better with sensor failures.
>
> Adjusting the reward function to narrow the gap between our approach and the PPO-MLP method under complete observations is a perfectly natural idea, but it bears little relevance to our objectives, as we do not wish for well-tuned or special reward functions to dominate the robot's performance, so we did not conduct related experiments.
>
> ## Hyperparameters tuning(for Weakness 4)
>
> The hyperparameter tuning is always necessary for RL methods, as well as gradient-based methods. Wrong hyperparameters can also cause the issue you mentioned in Weakness 4. To mitigate the problem, gradient-based optimisers such as Adam adopted sophisticated step adaptation and momentum. We have not engineered the model for performance in a similar direction yet, but we believe it is possible as future work. We tested the sensitivity of the parameters, and the results will be shown in the next comment.
>
> ## Regarding reward hacking (for Question 1)
> No, the robot controlled by the CPG-based networks performs naturally without exploiting loopholes. In comparison, the CPG trained by PPO uses quick oscillation to move, which is not a natural gait.
>
>
> ## Noise in the Reward (for Question 2)
>
> The DS is not sensitive to the noise in the reward, as it learns by exploring the parameter space and approaching where the reward is more likely to be higher. The shift of the oscillation center depends on many samples over many periods covering a range; hence, unbiased noise will be averaged to 0. There is also no deviation of the reward, so the DS is not sensitive to noise in the reward.
>
>
> ## Regarding the reward terms (for Question 4)
> The reward function employed in our work is the built-in function based on prior knowledge from IsaacLab. The team responsible for developing and maintaining IsaacLab is highly professional, ensuring these functions possess strong generalisation capabilities and can be adapted for use on other legged robots.

---

> > ### Author Response · Authors · 2025-11-25
> >
> > ## Regarding the reward efficiency (for Question 5)
> > In the experiments presented in the paper for optimising the **CPGs**, the GA and PPO algorithms failed to achieve training convergence because of dramatic changes in the CPG's dynamics and gradient issues, respectively. Hence, the assumptions that ‘both are stable’ and ‘excluding gradient issues’ prove unfounded. They cannot reach the target reward, let alone compare reward efficiency.
> >
> > ## Regarding the non-periodic tasks (for Question 6)
> > It is an instructive suggestion, thank you! In this work, we focus on the performance controlled by CPG, which is responsible for stereotyped motor behaviors, to test our method. Consequently, we have not yet attempted non-periodic tasks due to the limitation of CPG. To date, no one has publicly demonstrated the successful application of DS optimiser for non-periodic motion control. However, inspired by your suggestion, we shall undertake verification of preliminary evidence for its applicability to non-periodic tasks in our future work. The primary adjustments will concern the network architecture optimised by DS, so as the way to process reward, such as a critic converges on a goal to subgoals.
> >
> >
> > Thank you again for your comments and suggestions. Hope the above response addresses your concerns, and we look forward to your further feedback.

---

> > > ### Author Response · Authors · 2025-11-25
> > >
> > > ## Hyperparameter sensitivity
> > >
> > > To test this, we conduct ablation studies on six hyperparameters by comparing smaller/larger values against the original values used in our experiments. The results are analyzed as follows:
> > >
> > > 1. amp_init（Amplitude Initialization）
> > > This parameter affects the instantaneous weight of initialization as well as the early amplitude.
> > > Larger values enhance exploration, but this can lead to a more drastic decay of early rewards.
> > > Smaller values lead to a more stable uptrend, but there is a chance of getting stuck in the local optimal solution in the later stages of training.
> > > According to our experimental results, generally speaking, as long as it is not a too large or too small value, it will not have a significant impact on the final reward.
> > >
> > >     | Sample point | 0 | 1.25e7 | 2.5e7 | 3.75e7 | 5e7 | 6.25e7 | 7.5e7 | 8.75e7 | 1e8 |
> > >     | ---: | ---: | ---: | ---: | ---: | ---: | ---: | ---: | ---: | ---: |
> > >     | smaller |  | 16.0 | 19.14 | 20.04 | 18.65 | 21.53 | 20.37 | 22.36 | 22.88 |
> > >     | regular |  | 16.34 | 18.2 | 18.89 | 20.3 | 20.35 | 20.95 | 21.33 | 22.11 |
> > >     | bigger | -0.06 | 7.56 | 16.51 | 19.12 | 9.86 | 18.55 | 18.28 | 18.15 | 18.95 |
> > >
> > > 2. amp_update_rate (Amplitude Update Rate)
> > > This parameter affects the changing rate of amplitude, which in turn affects the convergence of dynamic synapses.
> > > Larger values lead to premature convergence and failure to discover the global optimum;
> > > Smaller values may enhance the robustness of the model, but risk unstable late-stage training, which will lead to convergence difficulties.
> > >
> > >     | Sample point | 0 | 1.25e7 | 2.5e7 | 3.75e7 | 5e7 | 6.25e7 | 7.5e7 | 8.75e7 | 1e8 |
> > >     | ---: | ---: | ---: | ---: | ---: | ---: | ---: | ---: | ---: | ---: |
> > >     | smaller |  | 15.77 | 18.62 | 19.94 | 20.08 | 21.25 | 21.05 | 20.68 | 21.95 |
> > >     | regular |  | 16.34 | 18.2 | 18.89 | 20.3 | 20.35 | 20.95 | 21.33 | 22.11 |
> > >     | bigger |  | 15.56 | 18.31 | 18.48 | 18.42 | 18.45 | 18.55 | 18.45 | 18.46 |
> > >
> > > 3. lr (Learning Rate)
> > > This parameter affects the changing rate of the weight center and amplitude. It is a parameter that regulates the overall rate of change. Therefore, too large or too small a parameter may have the same effect as the above two parameters. However, in general, if the other parameters are set properly, slight changes (we tried to enlarge it by twice or reduce it by half) will not affect the final result.
> > >
> > >     | Sample point | 0 | 1.25e7 | 2.5e7 | 3.75e7 | 5e7 | 6.25e7 | 7.5e7 | 8.75e7 | 1e8 |
> > >     | ---: | ---: | ---: | ---: | ---: | ---: | ---: | ---: | ---: | ---: |
> > >     | smaller |  | 13.29 | 18.84 | 20.01 | 20.52 | 20.86 | 21.03 | 21.15 | 21.58 |
> > >     | regular |  | 16.34 | 18.2 | 18.89 | 20.3 | 20.35 | 20.95 | 21.33 | 22.11 |
> > >     | bigger |  | 17.09 | 17.64 | 17.93 | 20.08 | 20.27 | 20.67 | 21.16 | 21.54 |
> > >
> > > 4. period (Oscillation Period)
> > > This parameter controls the normal mean used to generate the fluctuation period of dynamic synapses, which in turn affects the exploration of parameter space and whether the synapses have "effective" learning. In other words, it is probably the most important of these six parameters.
> > > If the period is too small, that is, the dynamic synapses fluctuate rapidly, then when the synapses obtain the reward caused by the current action, the weights may have moved away from the state that caused the action due to fluctuation, and then the changes of the fluctuation centers will not match the changes of the instantaneous weights, resulting in wrong learning.
> > > If the period is too large, the too slow fluctuations may lead to a local optimum. Take the walking task as an example, if a fluctuating period can allow the robot to complete dozens of behavioral steps, then the slight changes of weights in one behavioral step will result in limited exploration and thus risk falling into a local optimum.
> > > Therefore, we need to point out that this parameter should be estimated according to the tasks (but this is usually not difficult). For example, for periodic tasks, it is recommended that a fluctuation period should allow the robot to complete 2 - 4 times of behaviors. Sincerely thanks again for your reminder, we will supplement this section to the Discussion section of the paper.
> > >
> > >     | Sample point | 0 | 1.25e7 | 2.5e7 | 3.75e7 | 5e7 | 6.25e7 | 7.5e7 | 8.75e7 | 1e8 |
> > >     | ---: | ---: | ---: | ---: | ---: | ---: | ---: | ---: | ---: | ---: |
> > >     | smaller |  | 0.3 | 0.82 | 0.46 | 0.35 | 0.31 | 0.37 | 0.34 | 0.29 |
> > >     | regular |  | 16.34 | 18.2 | 18.89 | 20.3 | 20.35 | 20.95 | 21.33 | 22.11 |
> > >     | bigger | -0.06 | 16.92 | 17.78 | 18.55 | 19.24 | 19.7 | 20.27 | 19.92 | 20.42 |

---

> > > > ### Author Response · Authors · 2025-11-25
> > > >
> > > > 5. period_var (Oscillation Period Variance)
> > > > This parameter controls the normal variance used to generate the fluctuation period of dynamic synapses.
> > > > Large values will lead to some too large/small periods generated during the exploration process, which may have the same consequences explained above in the parameter period.
> > > > Small values will lead to almost constant periods, limit exploration diversity, and thus increase the probability of falling into a local optimum.
> > > >
> > > >     | Sample point | 0 | 1.25e7 | 2.5e7 | 3.75e7 | 5e7 | 6.25e7 | 7.5e7 | 8.75e7 | 1e8 |
> > > >     | ---: | ---: | ---: | ---: | ---: | ---: | ---: | ---: | ---: | ---: |
> > > >     | smaller |  | 0.42 | 1.51 | 3.34 | 2.83 | 0.96 | 5.01 | 1.51 | 3.18 |
> > > >     | regular |  | 16.34 | 18.2 | 18.89 | 20.3 | 20.35 | 20.95 | 21.33 | 22.11 |
> > > >     | bigger | -0.03 | 9.84 | 17.2 | 17.18 | 18.25 | 18.35 | 3.7 | 18.28 |  ||
> > > >
> > > > 6. weight_centre_update_rate (Weight Center Update Rate)
> > > > This parameter controls the changing rate of the center of the fluctuation in dynamic synapses.
> > > > Large values help the model quickly find a state better than the initial one in the early stage, but in the middle and late stages of training, it will lead to more drastic mutations of the actions, reduce the stability of the robot during exploration, and even lead to the inability to converge and training collapse.
> > > > Small values, on the contrary, will lead to inefficient training or fall into the local optimal value and converge prematurely. In fact, this parameter is more similar to learning_rate in training general artificial neural networks.
> > > >
> > > >     | Sample point | 0 | 1.25e7 | 2.5e7 | 3.75e7 | 5e7 | 6.25e7 | 7.5e7 | 8.75e7 | 1e8 |
> > > >     | ---: | ---: | ---: | ---: | ---: | ---: | ---: | ---: | ---: | ---: |
> > > >     | smaller |  | 15.72 | 18.69 | 18.73 | 19.63 | 19.96 | 20.17 | 20.37 | 20.5 |
> > > >     | regular |  | 16.34 | 18.2 | 18.89 | 20.3 | 20.35 | 20.95 | 21.33 | 22.11 |
> > > >     | bigger |  | 3.48 | 4.07 | 0.02 | 9.01 | 10.35 | 13.98 |  |  |
> > > >
> > > > In terms of the robustness to reward, this is not the primary focus of our current work. But it is of great significance; we shall validate it in future work.
> > > >
> > > > In terms of the robustness to reward, this is not the primary focus of our current work. But it is of great significance; we shall validate it in future work.

---

### Meta-Review · Area_Chair_TkwQ · 2026-01-16

**Summary:**

This paper proposes a bio-inspired gradient-free learning framework using a Dynamic Synapse (DS) optimizer to train Central Pattern Generator (CPG)-based neural networks for quadrupedal robot locomotion. The key contribution is enabling training of recurrent neural networks that typically suffer from gradient vanishing/explosion when using standard gradient-based methods like PPO.

The paper initially received three rejects (two strong). While the reviewers acknowledged the novelty of the approach and its ability to train architectures where PPO fails due to gradient instability, significant concerns were raised about: (1) limited practical advantages over standard RL approaches, (2) performance gap compared to PPO-trained MLPs, (3) insufficient comparisons with other gradient-free methods like CMA-ES, (4) validation limited to basic locomotion tasks, and (5) unclear scalability to more complex scenarios.

The authors responded with hyperparameter ablations and claimed improved parallelization of their method. However, they failed to respond to follow up questions from the reviewers. While two reviewers did not respond to the rebuttal, I suspect none of the reviewers would have been strongly convinced by rebuttal to recommend acceptance.

**Reviewer Concerns:**

Significant concerns were raised about: (1) limited practical advantages over standard RL approaches, (2) performance gap compared to PPO-trained MLPs, (3) insufficient comparisons with other gradient-free methods like CMA-ES, (4) validation limited to basic locomotion tasks, and (5) unclear scalability to more complex scenarios. These were not addressed by the rebuttal.

**Reviewer Scores:**

While two reviewers did not respond to the rebuttal, I suspect none of the reviewers would have been strongly convinced by rebuttal to recommend acceptance.

---

### Decision · Program_Chairs · 2026-01-26

Reject